



# Effects of the arrival of fresh organic matter on eroded and nutrient-depleted trawling grounds (Gulf of Castellammare, SW Mediterranean)

Sarah Paradis[1], Antonio Pusceddu[2], Pere Masqué[1,3,4,5], Pere Puig[6], Davide Moccia[2], Tommaso Russo[7], Claudio Lo Iacono[6,8]

[1]Institut de Ciència i Tecnologia Ambientals, Universitat Autònoma de Barcelona, Bellaterra, 08193, Spain
[2]Dipartimento di Scienze della Vita e dell'Ambiente, Università degli Studi di Cagliari, Cagliari, 09126, Italy
[3]Departament de Física, Universitat Autònoma de Barcelona, Bellaterra, 08193, Spain
[4]School of Natural Sciences, Centre for Marine Ecosystems Research, Edith Cowan University, Joondalup, WA 6027, Australia
[5]School of Physics and Oceans Institute, University of Western Australia, Crawley, WA 6009, Australia
[6]Marine Sciences Institute, Consejo Superior de Investigaciones Científicas, Barcelona, 08003, Spain
[7]Laboratory of Experimental Ecology and Aquaculture, Department of Biology, University of Rome Tor Vergata, Rome, 00133, Italy
[8]National Oceanography Centre, University of Southampton Waterfront Campus, Southampton, SO14 3ZH, United Kingdom

*Correspondence to*: Sarah Paradis (sarah.paradis@uab.cat)

**Abstract.** Bottom trawling in the deep sea is one of the main drivers of sediment resuspension, eroding the deep seafloor and altering the content and composition of sedimentary organic matter (OM). The physical and biogeochemical impacts of bottom trawling on the seafloor were studied in the continental slope of the Gulf of Castellammare, Sicily (Southwestern Mediterranean) through the analysis of two triplicate sediment cores collected in trawled and untrawled sites (~550 m water depth) during the summer of 2016. Geochemical and sedimentological parameters (excess $^{210}$Pb, excess $^{234}$Th, $^{137}$Cs, dry bulk density, and grain size), elemental (organic carbon and nitrogen) and biochemical composition of sedimentary OM (proteins, carbohydrates, lipids), as well as its freshness (phytopigments) and degradation rates were determined in both coring locations. The untrawled site had a sedimentation rate of 0.15 cm yr$^{-1}$ and presented a 6-cm thick surface mixed layer that contained coarser sediment with low excess $^{210}$Pb concentrations, possibly resulting from the resuspension, posterior advection, and eventual deposition of siltier and older sediment from adjacent trawling grounds. In contrast, the trawled site was characterized by highly eroded and compacted century-old sediment, as shown by the lack of excess $^{210}$Pb and high dry bulk densities. The continuous erosion in the trawled site has led to the depletion of OM, which were between 20% and 60% lower than those in the untrawled site, as well as to statistically significant differences in the biochemical composition of OM. Nevertheless, the upper 2 cm of the trawled site consisted of recently accumulated sediments, enriched in excess $^{234}$Th, excess $^{210}$Pb, and phytopigments, which had similar OM contents to surface sediments from the untrawled core. The arrival of fresh sediment in a chronically-trawled deep-sea site that is generally deprived of OM was associated with an enhancement of remineralization rates, reflected by protein turnover rates of 0.025 d$^{-1}$, which doubled the rates quantified in surface sediments of the untrawled site. We conclude that the detrimental effects of bottom trawling can be temporarily and partially abated by the arrival of fresh

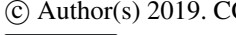



and nutritionally-rich OM, which stimulate the response of benthic communities. However, these ephemeral deposits are likely to be swiftly eroded due to the high trawling frequency over fishing grounds, highlighting the importance of establishing management strategies to mitigate the impacts of bottom trawling.

# 1 Introduction

Bottom trawling is among the most extensive forms of anthropogenic activities affecting marine ecosystems (Amoroso et al., 2018; Eigaard et al., 2017) and it is one of the most harmful in terms of fish stock overexploitation (Pauly et al., 2002), destruction of habitats (Kaiser et al., 2002; Simpson and Watling, 2006), and physical impact it exerts on the sediments (Martín et al., 2014a; Oberle et al., 2018; Puig et al., 2012). Since bottom trawling targets benthic and demersal fisheries, its gear is designed to be in continuous contact with the seafloor, scraping the bottom, resuspending large volumes of sediment (O'Neill

and Ivanović, 2016; Palanques et al., 2014), and causing significant erosion (Martin et al., 2014b; Oberle et al., 2016). The resuspension of sediment releases nutrients and organic matter to the overlying water column, and degradation of sedimentary organic matter can be accelerated through enhanced microbial activity (Durrieu de Madron et al., 2005; Pusceddu et al., 2005b, 2015). Additionally, concentrations of sedimentary organic matter in superficial sediments tend to increase (Palanques et al., 2014; Pusceddu et al., 2005a), possibly as the result of mixing and oxygenation of the deeper layers by the trawling gear, which

ultimately stimulate mineralisation of buried and refractory organic matter (Polymenakou et al., 2005; van de Velde et al., 2018). Most of these impacts have been documented in shallow environments, where sediment and organic matter fluxes are generally high and sediment resuspension and organic matter remineralization induced by bottom trawling can be comparable to those induced by natural high-energy events such as storms (Buscail et al., 1990; Dellapenna et al., 2006; Durrieu de Madron et al., 2005; Pusceddu et al., 2005b). Since these natural physical disturbances are persistent, shallow benthic communities

generally present higher resilience to the impacts of bottom trawling than communities that live in less disturbed areas, such as the deep sea (Kaiser, 1998).

However, bottom trawling has been progressively expanding to deeper environments (>200 m depth) over the last 60 years (Morato et al., 2006; Roberts, 2002), driven by technological advances in parallel with an on-going depletion of shallow-water fisheries (Koslow et al., 2000; Martín et al., 2014a). At such depths, natural sediment fluxes to the seafloor and resuspension

processes tend to be low. Hence, bottom trawling has become a major mechanism of sediment resuspension on continental slopes, leading to eroded environments in trawling grounds (Martín et al., 2014b, 2014c; Puig et al., 2012). Resuspended particles can then be exported by ambient currents across- and along-margin as enhanced nepheloid layers (Arjona-Camas et al., 2019; Wilson et al., 2015), ultimately generating anthropogenic sedimentary depocenters (Puig et al., 2015; Paradis et al., 2017, 2018).

Contrary to the observed increases in total organic carbon on continental shelf trawling grounds (Palanques et al., 2014; Polymenakou et al., 2005; Pusceddu et al., 2005a), the continuous removal of sediment by trawlers on continental slopes of this region have significantly impoverished bulk organic carbon as well as its labile and fresh pools (Martín et al., 2014b;

**Biogeosciences** Open Access
Discussions
EGU

Pusceddu et al., 2014, Sañé et al., 2013). The loss of organic matter has also reduced organic carbon turnover rates on trawling grounds, severely impacting the meiofauna and, at the same time, promoting the abundance of taxa with opportunistic life strategies (Pusceddu et al., 2014). However, the combined effect of bottom trawling erosion along with the alteration of sedimentary organic matter, which usually represents the fundamental energy source for commercial deep-sea benthic species,

is not fully understood.

The Gulf of Castellammare holds one of the most important bottom trawling grounds in the Northern Sicilian shore (Southwestern Mediterranean Sea). Fishing stocks within this gulf were declining alarmingly until the Sicilian Government established in 1990 a trawling ban area in the inner shelf, delimited by the junction between Capo Rama and Torre dell'Uzzo (Fig. 1). Since the establishment of this closure, both demersal biomass and catch per unit effort (CPUE) of artisanal fisheries

(non-towed bottom gear and pelagic gear) have increased in that area (Pipitone et al., 2000; Whitmarsh et al., 2002). However, bottom trawlers have been concentrating their efforts beyond the restricted area in the mid-continental slope (> 500 m depth), leading to a decrease in CPUE since the trawl ban as a result of the continuous overexploitation of fishing stocks (Arculeo et al., 2014; Whitmarsh, 2002).

Despite the numerous studies that address the effects of the trawl ban in the Gulf of Castellammare (Fanelli et al., 2008;

Romano et al., 2016; Pipitone et al., 2000; Whitmarsh et al., 2002), no studies have yet assessed the impacts of bottom trawling on the Gulf's sedimentary environment. This study aims to reveal whether erosion prevails in bottom trawling grounds and what are the consequent alterations on sedimentary organic matter by comparing sediment cores collected in a trawled and untrawled site in the Gulf of Castellammare. The degree of erosion will be estimated based on sedimentological parameters and radioactive tracers with different half-lives ($^{210}$Pb, $t_{1/2}$ = 22.3 years; $^{234}$Th, $t_{1/2}$ = 24.1 days), whereas the alterations on

sedimentary organic matter will be determined based on its quantity, composition, and nutritional quality. The coupled analyses of radioactive tracers and biomarkers will also provide insights on the effects of the arrival of fresh sediment on impacted trawling grounds.

## 2 Methods

### 2.1 Study area

The Gulf of Castellammare is one of the widest bays of the northern coast of Sicily, with over 70 km of coastline, enclosed by the Cape Rama to the East and Cape San Vito to the West (Fig. 1). An easterly anticyclonic current dominates the Gulf's regional circulation (Istituto Idrografico della Marina, 1982). The seafloor morphology consists of a sub-horizontal gently sloping continental shelf that extends approximately 5 km offshore. The continental slope is around 11° steep down to 500 m water depth, and then gradually decreases to around 1.5° at 1300 m water depth (Lo Iacono et al., 2014). Several small, narrow

submarine canyons cut the slope, breaching the shelf break at 120 to 140 m depth (Lo Iacono et al., 2014). Small seasonal torrents discharge into the Gulf, namely the Nocella, Jato and San Bartolomeo rivers, with annual average discharges between





0.24 m³·s⁻¹ and 0.32 m³·s⁻¹ (Regione Siciliana, 2007). Storm-induced flash floods can cause short flushing events of up to 1.2 m³·s⁻¹ that transport significant amounts of nutrients into the sea (Calvo and Genchi, 1989).

## 2.2 Sediment core sampling

In the framework of the FP7 EU-Eurofleets 2 ISLAND (ExplorIng SiciLian CAnyoN Dynamics) cruise on board the R/V
Angeles Alvariño, sediment cores were collected in August 2016 from trawled and untrawled sites in the Gulf of Castellammare. Sampling locations were selected based on the distribution of operating trawlers using data from Vessel Monitoring System (VMS) (see Sect. 2.8).

A total of five multicore deployments were conducted using a K/C Denmark A/X six-tube multicorer (inner diameter 9.4 cm) in trawled and untrawled sites along the 550 m contour lines. However, only one trawled and one untrawled sites could be
sampled (Fig. 1), possibly due to high sediment compaction at the trawled sites, as experienced by Martin et al. (2014b), and/or due to the swell during the coring operation that could hamper a successful triggering of the multicorer.

Triplicate sediment cores were retrieved at each site from three independent multicore deployments to account for spatial variability. The sediment cores were sliced on deck (0-1, 1-3, 3-5, 5-7, and 7-9 cm) and stored in calcinated aluminium foil at -20°C until analysis. At each site, a sediment core from one of the three deployments was reserved for sedimentary,
radiochemical, and elemental analyses. This sediment core was sliced on-deck at 1 cm intervals and kept in sealed plastic bags at -20°C until freeze-dried in the laboratory for analyses.

Prior to sediment recovery, the remotely operated vehicle (ROV) Seaeye *Falcon*, from the University of Plymouth (UK), collected visual evidence of trawling impact at the trawled sampling site and of no impact at the control site to corroborate the sampling strategy (Fig. S1).

## 2.3 Sedimentary characteristics

Dry bulk densities of sediment cores were calculated by dividing the net dry weight corrected for salt content by the sample volume. Grain size fractions were obtained using a Horiba Partica LA-950V2 particle-size analyser, with an accuracy of 0.6% and a precision of 0.1%. Prior to analysis, 1-4 g of sample was oxidized using 20% $H_2O_2$ and sediment particles were disaggregated with $P_2O_7^-$.

## 2.4 Radiochemical analyses

Concentrations of ²¹⁰Pb were determined through the analysis of its decay product ²¹⁰Po by alpha spectrometry following the method described by Sanchez-Cabeza et al. (1998), assuming secular equilibrium of both radionuclides at the time of analysis. Between 150 and 300 mg of homogenized ground samples were spiked with ²⁰⁹Po as a chemical yield and microwave-digested using concentrated $HNO_3$, HF, and $HBO_3$. The resulting solutions were evaporated and reconditioned with 1 M HCl. Polonium
isotopes were spontaneously deposited onto silver discs while stirring at 70°C for at least 8 hours. Alpha emissions of ²⁰⁹Po (4883 keV) and ²¹⁰Po (5304 keV) were quantified using Passivated Implanted Planar Silicon (PIPS) detectors (CANBERRA,





Mod. PD-450.18 A.M.) and the Genie™ data acquisition software. Supported concentrations of $^{210}$Pb in the sediment cores were obtained by averaging constant concentrations of total $^{210}$Pb from the bottom of the core, assuming complete decay of excess $^{210}$Pb at this depth. Supported $^{210}$Pb concentrations were corroborated by measuring $^{226}$Ra concentrations through its decay product $^{214}$Pb (295 and 352 keV) in several samples along each core by gamma-spectroscopy, using calibrated

geometries in a well-type high-purity germanium detector (CANBERRA, Mod. GCW3523).

Concentrations of $^{234}$Th were also measured by gamma-spectroscopy through the 63 keV emission line. Given its short half-life (24.1 days), samples were measured within two half-lives (~6 weeks) since sampling, which only allowed the measurement of the upper 5 cm of the trawled and untrawled cores. Samples were re-measured at least 6 months later, after excess $^{234}$Th had decayed, to obtain supported $^{234}$Th concentrations, equivalent to $^{238}$U concentrations. Excess $^{234}$Th was calculated by

subtracting total $^{234}$Th from supported $^{234}$Th, accounting for $^{234}$Th decay and in-growth from $^{238}$U since sampling.

Concentrations of $^{137}$Cs were also quantified by gamma-spectroscopy through the emission line at 662 keV. Gamma measurements of the untrawled sediment core were extended in depth to 20 cm, whereas in the trawled core measurements were limited to the upper 5 cm.

**2.5 Elemental analyses**

Analyses of total carbon, organic carbon (OC) and total nitrogen (TN) were carried out with an elemental analyser (Costech ECS Analyzer 4010), according to the procedure described in Nieuwenhuize et al. (1994). Samples for OC analysis were first decarbonated by acid-fuming the samples in the presence of 12 N HCl during 24 h and repeatedly adding 100 µL of 2 N HCl to the sample until effervescence ceased. Inorganic carbon (IC), quantified as the difference between total carbon and organic carbon, was converted to calcium carbonate ($CaCO_3$) concentrations using the molecular mass ratio of $CaCO_3$:IC (100/12),

assuming all inorganic carbon present is in the form of $CaCO_3$. To account for analytical error, replicate analyses were performed for samples every 5 cm throughout the cores. An average percentage error of 1.2 % was obtained for carbon whereas nitrogen presented a slightly higher average percentage error of 1.9 %.

**2.6 Biochemical composition of sedimentary organic matter**

Total proteins, carbohydrates, and lipids were quantified spectrophotometrically (Varian Cary® 50 UV-Vis) according to the

methods described in Hartree (1972) and modified by Rice (1982), Gerchakov and Hatcher (1972), Bligh & Dryer (1959), and Marsh and Weinstein (1966). The analyses of proteins and lipids were carried out on 0.1-0.6 g of frozen sediment, whereas carbohydrate analyses were done on previously-dried sediment. Protein, carbohydrate and lipid contents were transformed into carbon equivalents using 0.49, 0.4 and 0.75 mg C·mg$^{-1}$ as conversion factors, respectively, and their sum reported as biopolymeric C (Fabiano et al., 1995). Chlorophyll-a and phaeopigments, after extraction with 90% acetone, were quantified

fluorometrically (Shimadzu RF-6000) according to Lorenzen and Jeffrey (1980), and modified by Danovaro (2010) for sediments. Total phytopigment concentrations were defined as the sum of chlorophyll-a and phaeopigment concentrations and converted into carbon equivalents using a conversion factor of 40 (Pusceddu et al., 2010).





## 2.7 Sedimentary OM freshness and degradation rates

The contribution of phytopigment to biopolymeric C was used as a proxy to estimate OM freshness: since in the deep sea there is no *in situ* primary production, higher values of this ratio are associated with recently deposited material of algal origin (Pusceddu et al. 2010).

Since N is the most limiting factor for heterotrophic nutrition and proteins are N-rich products, sedimentary OM degradation was estimated using the degradation rate of proteins, obtained from the analysis of extracellular aminopeptidase activities. Aminopeptidase activity was estimated fluorometrically after incubation of approximately 0.1 g of sediment with 100 µM L-leucine-4-methylcumarinyl-7-amide for 1 h in the dark. This substrate, when exposed to extracellular aminopeptidase, produces fluorescence with an intensity proportional to the enzyme activity. Fluorometric analyses were carried out before and

after incubation, and the difference was used to calculate protease activities (Danovaro, 2010). The results were converted to carbon equivalents using the conversion factor of 72 ng C·nmol protease$^{-1}$ (Fabiano and Danovaro, 1998). Turnover rates were then calculated by dividing protein-C degradation rates by protein-C sedimentary contents.

## 2.8 Trawling effort from VMS data

Fishing intensity of the Italian bottom trawling fleet was obtained using data provided by Vessel Monitoring System (VMS),

the main tracking device used for monitoring fishing activities. According to the Common Fisheries Policy of the European Union (European Commission, 2003), fishing vessels with length-over-all equal to or larger than 15 m must be equipped with a VMS trasmittant, called "Blue-Box". It estimates the position of the vessel by Global Positioning System, and sends this information, along with the speed and heading of the vessel, to the network of the Coastal Guard by Inmarsat-C to the Fishing Monitoring Centre in less than 10 min at 2-hour time intervals. Fishing intensity was calculated using yearly VMS data from

2007 to 2015, whereas for 2016, only VMS data from January 1$^{st}$ to August 10$^{th}$ were taken into account, prior to sampling. Trawling frequency was represented as number of times trawled per grid cell (200 x 200 m) during each year. The native VMS data were processed using the R package VMSbase (Russo et al., 2014). The size of the grid was defined considering the error associated to the reconstruction of the trawling hauls as described by Russo et al. (2011).

## 2.9 Statistical analyses

Statistical analyses were used to test whether OM quantity and biochemical composition (protein, carbohydrate, lipid, and phytopigment concentrations), freshness (the phytopigment to biopolymeric C ratio), and degradation rates (sedimentary protein turnover rates) were statistically different between trawled and untrawled sites in the upper 9 cm of sediment cores. The analysis consisted of two orthogonal factors: site (trawled vs. untrawled) and depth in the sediment (5 levels: 0-1 cm, 1-3 cm, 3-5 cm, 5-7 cm, 7-9 cm). Permutational analyses of variance (PERMANOVA), either in the univariate (variable by

variable) or multivariate contexts, were based on Euclidean distances of previously normalized data using 999 permutations of residuals with unrestricted permutation of raw data (univariate tests) or under a reduced model (multivariate tests) (Anderson



2001). Since for almost all tests the interaction between factors was significant, we conducted post-hoc permutational pairwise comparison tests between trawled and untrawled sites for each sediment layer and among sediment layers for trawled and untrawled sites, separately. Given the restricted number of unique permutations, p-values were obtained from Monte Carlo simulations (Anderson and Robinson, 2003). Bi-plots produced after Canonical Analysis of Principal components (CAP) were used to visualize the differences between trawled and untrawled samples in terms of organic matter biochemical composition (Anderson and Willis, 2003). All statistical analyses were performed using the routines included in the PRIMER 6+ software.

## 3 Results

Bottom trawling in the Gulf of Castellammare is limited to the mid-slope (> 500 m), beyond the trawling-ban area. In the main bottom trawling ground, where the trawled core was retrieved, hauls generally follow the contour lines on a W-E direction (Fig. 1). Fishing effort generally increased since 2007, with a predominating trawling frequency of 1 to 40 hauls per grid cell during that year, which then increased to around 60 to 100 hauls per grid cell since 2013 (Fig. S2). A smaller trawling ground towards the eastern side of the Gulf, close to Cape Rama, opened since 2012 (Fig. 1, S2). Although trawling frequency for 2016 was only computed from January to August, this year presented higher fishing effort in comparison to the previous years.

### 3.1 Physical characteristics

The untrawled sediment core presented an excess $^{210}$Pb concentration profile that extended to 25 cm in depth, with a total inventory of $17900 \pm 900$ Bq·m$^{-2}$ (Fig. 2a; Table 1). In the upper 6 cm, excess $^{210}$Pb concentrations slightly decreased towards the surface from $372 \pm 22$ Bq·kg$^{-1}$ to $272 \pm 15$ Bq·kg$^{-1}$ (Fig. 2a). Below, excess $^{210}$Pb concentrations presented a continuous decrease between 6 and 25 cm, from which an average sediment accumulation rate of $0.090 \pm 0.003$ g·cm$^{-2}$·yr$^{-1}$, equivalent to $0.151 \pm 0.005$ cm·yr$^{-1}$ ($R^2 = 0.995$) (Table 1) was calculated applying the Constant Flux : Constant Sedimentation model (CF:CS, Krishnaswamy et al., 1971). This sedimentation rate was independently validated by $^{137}$Cs. Detectable concentrations of $^{137}$Cs appeared at 17 cm depth, which was ascribed to the first detonations of thermonuclear weapons in early 1950s. Above, $^{137}$Cs concentrations depicted a broad concentration maximum at 8-13 cm, centred at 10-11 cm. This maximum was attributed to the combined accumulation of the maximum fallout prior to the cessation of nuclear atmospheric testing in 1963 as well as the deposition of $^{137}$Cs emitted from the Chernobyl accident in 1986 (Fig. 2a). The deposition of $^{137}$Cs from each of these events could not be distinguished due to the low sedimentation rate and the sampling resolution of this core. Concentrations of excess $^{234}$Th ranged between 71 and 97 Bq·kg$^{-1}$ between 2 and 5 cm, decreased to $37 \pm 7$ Bq·kg$^{-1}$ at 1-2 cm and was not detected on surface sediments (0-1 cm) of the core (Fig. 2b). The penetration depth of excess $^{234}$Th could be greater than the upper 5 cm analysed, leading to an inventory of at least 1080 Bq·m$^{-2}$ (Table 1). Dry bulk density of the untrawled site remained constant in the upper 7 cm at ~0.5 g·cm$^{-3}$, gradually increased to ~0.7 g·cm$^{-3}$ at 15 cm, and then remained relatively constant with depth (Fig. 2c). The upper 6 cm presented coarser grain size consisting of higher silt (77 %) and lower clay (22 %) fractions, in comparison to the rest of the core, which had lower silt (44 %) and higher clay (54 %) fractions (Fig. 2c, Table





1). CaCO$_3$ concentrations were constant in the upper 10 cm, with an average concentration of 18.4 ± 0.4 %, decreased to ~15 % at 15-20 cm, and then slightly increased with depth until ~20 % at 30 cm (Fig. 4d).

In the sediment core collected from the trawled site, both excess $^{210}$Pb and excess $^{234}$Th were only present in the upper 2 cm, with inventories of 340 ± 30 Bq·m$^{-2}$ and 1000 ± 50 Bq·m$^{-2}$, respectively (Table 1, Fig. 3a,b). $^{137}$Cs was not detected in the

upper 5 cm analysed. Dry bulk density rapidly increased from 0.4 g·cm$^{-3}$ to 0.7 g·cm$^{-3}$ in the upper 3 cm and kept increasing to ~0.8 g·cm$^{-3}$ at 13 cm, from where it remained relatively constant with depth (Fig. 3c). Despite the slight coarsening of grain size in the upper 2 cm, this core presented constant grain size along the upper 37 cm, consisting of higher clay (57 %) and lower silt (41 %) fractions, (Fig. 3c, Table 1), similar to grain sizes below 6 cm of the untrawled site. Concentrations of CaCO$_3$ averaged at 26.9 ± 0.7 % in the upper 10 cm, whereas the deeper layers had lower concentrations of ~24 % (Fig. 3d).

**3.2 Sedimentary organic matter**

Trawled and untrawled sediment cores presented different OC and TN concentration profiles (Fig. 4). The untrawled core had OC concentrations of ~0.9 % in the upper 20 cm which then decreased to ~0.8 % at 35 cm. In contrast, the trawled core presented OC concentrations that fluctuated between ~0.8 % and ~0.7 % with depth. Similarly, concentrations of TN in the untrawled core were ~0.13 % in the upper 20 cm, which then decreased to ~0.10 % at 35 cm, whereas the trawled core presented

TN concentrations varying between ~0.11 % and ~0.09 % with depth. For both OC and TN, concentrations in the upper 20 cm were ~20 % lower in the trawled site in comparison to the untrawled site, reaching similar concentrations at 30 cm in depth (Fig. 4). Profiles of OC/TN ratio presented similar values in both sites, varying between 6.5 and 7.5 throughout the core (Fig. 4c).

In general, organic matter quantity was higher in the untrawled site than in the trawled site, with the exception of surface

layers, which presented similar concentrations in both sites (Fig. 5). Protein concentrations in the untrawled site were 1.5 mg C·g$^{-1}$ in the upper 3 cm, increased to maximum concentrations of ~3 mg C·g$^{-1}$ at 3-7 cm, and decreased to 2.4 mg C·g$^{-1}$ at 7-9 cm. The trawled site had similar 1.5 mg C·g$^{-1}$ protein concentrations in surface sediment that increased to almost 2.5 mg C·g$^{-1}$ in the deepest layer analysed (Fig. 5a). Carbohydrate concentrations in the untrawled site presented decreasing concentrations from between 0.7 and 0.9 mg C·g$^{-1}$ in the upper 3 cm to 0.44 mg C·g$^{-1}$ in the deepest layer, whereas the trawled site had

constant concentrations of 0.42 mg C·g$^{-1}$ in the upper 9 cm of the trawled site (Fig. 5b). Untrawled and trawled sites had similar lipid concentrations of ~0.6 mg C·g$^{-1}$ in the topmost sediment layer, decreasing to 0.34 ± 0.03 mg C·g$^{-1}$ and 0.17 ± 0.01 mg C·g$^{-1}$ at 7-9 cm of the untrawled and trawled cores, respectively (Fig. 5c). Biopolymeric C concentrations were constant (2.5 ± 0.1 mgC·g$^{-1}$) along the upper 9 cm of the trawled core, whereas concentrations increased in the untrawled core from 2.9 ± 0.2 mg C·g$^{-1}$ in the topmost layer to ~4 mg C·g$^{-1}$ in the 3-7 cm layer, before returning to similar values observed in the surface

layer (Fig. 5d). In both sites, phytopigment profiles showed decreasing trends, with a sharper decrease in the trawled site from 126 ± 7 µgC·g$^{-1}$ in the top layer to relatively constant values of 39 ± 5 µgC·g$^{-1}$ below 3 cm, whereas the untrawled site showed a gradual decrease from 161 ± 18 µgC g$^{-1}$ in the topmost layer to 77 ± 12 µgC g$^{-1}$ in the deepest one (Fig. 5e).





Statistical analyses of the data indicated that both sampling site and core depth had significant effects on the quantity of sedimentary OM (Table S1). The post-hoc comparison tests (Table S2) demonstrated that OM contents between trawled and untrawled sites were generally statistically different only below the topmost sediment layer (0-1 cm), with the exception of carbohydrates.

## 3.3 Organic matter biochemical composition, freshness, and turnover rates

Variations in the biochemical composition of sedimentary OM in terms of protein, carbohydrate, lipid, and phytopigment contents were assessed through PERMANOVA test (Table S3). The results revealed that the biochemical composition of sedimentary OM not only differed between trawled and untrawled sites, but that these differences varied depending on the depth in the core. The consequent post-hoc pairwise tests between the trawled and untrawled sites at all sediment layers indicated that the biochemical composition of sedimentary OM, similarly to the OM content, varied significantly between trawled and untrawled sites at all depths, excluding surface sediments (Table S4). The bi-plot produced after the canonical analysis of principal components (Fig. 6) revealed that the vertical variations in the biochemical composition of sedimentary OM in the untrawled sediment core are greater than those observed at the trawled site; and that the biochemical composition of superficial sediments from the trawled site resemble that of superficial sediments from the untrawled site.

The contributions of phytopigment to biopolymeric C and the protein turnover rates in trawled and untrawled sites are given in Table S5 and illustrated in Fig. 7. The relative contribution of phytopigments to biopolymeric C were similar in superficial sediments of trawled and untrawled sites, with relatively high values (~5 %) that decreased with depth in both sites, although more pronouncedly in the trawled site than in the untrawled site (Fig. 7a). Turnover rates were significantly higher in the upper 3 cm of trawled site (0.017-0.025 $d^{-1}$) in comparison to the ~0.015 $d^{-1}$ of superficial sediment of the untrawled site (Table S5b, Fig. 7b). Below, turnover rates decreased to ~0.005 $d^{-1}$ for both sites.

## 4 Discussion

## 4.1 Long-term impacts of intense bottom trawling

Evidence of the long-term impacts of intense bottom trawling are clear in a trawled site of the Gulf of Castellammare. ROV images from the trawled area present a barren seafloor, with several deep linear furrows hundreds of meters long and up to 70 cm deep, presumably caused by heavy trawling gear (Fig. S1a). The shallow penetration depth of excess $^{210}$Pb in the trawled core suggests that only the upper 2 cm of sediment had been recently deposited on top of highly compacted sediments (0.7-0.8 g·$cm^{-3}$) that had accumulated more than 100 years ago, considering the absence of excess $^{210}$Pb (Fig. 3a,c). These sedimentological patterns are characteristic of a seafloor eroded by trawling activities, as observed in other trawled regions with shallow horizons of excess $^{210}$Pb and exposed over-consolidated, century-old sediments (Martín et al., 2014b; Oberle et al., 2016; Paradis et al., 2017, 2018). The uprooting of old sediment in this trawled site, with only a thin accumulation of recent sediment in superficial layers, reveals that the rate of sediment erosion induced by the high trawling frequency is greater than





sediment accumulation rates in the Gulf of Castellammare. Furthermore, the coincident penetration depth of both excess [210]Pb and [234]Th in the trawled site (Fig. 3a,b) indicates that the accumulation of fresh sediment would have occurred after the passage of the last trawler over the sampling area, and thus that trawling frequency controls the residence time of fresh sediment in these trawling grounds.

In contrast, excess [210]Pb inventories were two orders of magnitude higher in the untrawled core. This core had a sedimentation rate of $0.151 \pm 0.005$ cm·yr[-1], comparable to those quantified at similar depths in the Mediterranean Sea (DeGeest et al., 2008; Miralles et al., 2005; Sanchez-Cabeza et al., 1999). The upper 6 cm of this core had excess [210]Pb concentrations that slightly decreased towards the surface, a deep penetration depth of excess [234]Th, and low dry bulk densities (Fig. 2a,b), altogether signs of biological mixing (Arias-Ortiz, et al. 2018; Pope et al., 1996; van Weering et al., 1998). The influence of bioturbation in
this site is corroborated by the presence of several burrows directly observed during ROV dives prior to sampling (Fig. S1b). Sediment mixing caused by bioturbation could explain the broad [137]Cs concentration maximum observed at 8-13 cm, attributed to the combined accumulation of [137]Cs from the 1986 Chernobyl accident and from the 1963 global fallout, as well as diluting the 1954 signal with depth (Fig. 2a). However, bioturbation alone cannot account for the coarsening of sediment observed in the upper 6 cm (Fig. 2c; Table 1). This coarsening could probably be explained by the arrival of siltier (i.e., coarser) sediment
originating from trawling-induced resuspension at an adjacent trawling ground located ~1 km up-current from this sampled site, which would have been posteriorly redistributed along the margin and preserved within the surface mixed layer (Fig. 1). Excluding the fresh superficial layers of the trawled site, the chronic erosion induced by bottom trawling resuspension considerably depleted the trawled site of both OC and TN by ~20 % in comparison to the untrawled site (Fig. 4; Table 1). Concentrations of OC and TN were similar in the deeper layers of both sites, where sediments would have accumulated more
than a century ago, revealing that bottom trawling re-exposes old sediment impoverished in OM (Fig. 4; Table 1). Similarly, the trawled site had lower proteins (-5 to -38 %), carbohydrates (-13 to -58 %), lipids (-36 to -52 %), biopolymeric carbon (-12 to -37%), and phytopigments (-53 to -67 %) than the untrawled site, with the exception of superficial layers (Fig. 5). These results are in accordance with previous studies that showed comparable losses of organic matter in trawling grounds, reinforcing the concept that chronic and intensive bottom trawling depletes trawling grounds of sedimentary organic matter,
promoting the degradation of deep-sea sedimentary habitats (Martin et al., 2014b; Pusceddu et al., 2014; Sañé et al., 2013), whereas the opposite occurs in shallow shelf trawling grounds (Palanques et al., 2014; Pusceddu et al., 2005a).

### 4.2 Effects of the arrival of fresh sediment

During the ISLAND Cruise, ROV dives showed high settling fluxes of large particulate matter aggregates to the seafloor (Fig. S1). In both sampled sites, evidence of recent accumulation of surface sediments was provided by the presence of excess [234]Th
and high concentrations of phytopigments (Fig. 5e), a compound that usually represents the most important food source for deep-sea heterotrophic consumption (Pusceddu et al., 2010; Stephens et al., 1997). This indicates that the Gulf of Castellammare was receiving highly nutritious organic matter inputs during the sampling period. Indeed, both the composition of sedimentary organic matter and the relative contribution of phytopigment to biopolymeric C were similar in the fresh





superficial sediments in both trawled and untrawled sites. However, the subsurface, century-old sediments of the trawled site have distinctively different organic matter composition and significantly lower nutritional quality in comparison to its untrawled counterpart (Fig. 6, 7a). This suggests that, aside from the ephemeral deposition of fresh OM that will be swiftly eroded by bottom trawlers' gear, deep-sea trawling grounds are generally characterized of nutritionally-poor organic matter

contents (Pusceddu et al., 2014; Sañé et al., 2013), which increases the dependence on the supply of fresh OM in order to sustain benthic communities. This hypothesis is corroborated by the higher OM turnover rates in surface sediment of the trawled site in comparison to the untrawled site (Fig. 7b), which reveal a promptly-enhanced stimulation of microbial activities resulting from the recent accumulation of fresh and nutritionally-enriched OM. In fact, benthic communities in areas that have severe nutrient limitations, such as in eroded sediment of this trawled site or in oligotrophic deep-sea regions, react

instantaneously to food pulses (Bett et al., 2001; Fabiano et al., 2001; Witte et al., 2003). In contrast, the untrawled site, characterized of relatively higher total organic matter contents as well as fresh and bioavailable compounds throughout the core, presented slowly decreasing concentrations of phytopigment to biopolymeric C with depth as well as lower protein turnover rates, revealing a lower consumption of labile OM and a relatively reduced dependence on the arrival of fresh OM (Fig. 7a).

The higher OM turnover rates observed in the freshly-deposited surface layers of the trawled site also indicate a greater efficiency of OM consumption and remineralization in the trawling ground in comparison to the untrawled site. This enhanced mineralization through self-priming can occur when fresh and degradable organic matter, such as fresh phytoplankton, arrives to areas with more refractory compounds (Aller, 1994; Canfield, 1994; van Nugteren et al., 2009), as was observed in a shallow trawling-disturbed area in the southern North Sea, off the Belgian coast (van de Velde et al., 2018). Similarly, bottom trawling

in the shallow Thermaikos Gulf (Aegean Sea) intensified microbial activities, which enhanced nutrient cycling and organic carbon mineralisation (Polymenakou et al., 2005; Pusceddu et al., 2005a). This could have been attributed to the combined effect of trawling-induced mixing of superficial labile OM with more degraded subsurface OM, along with the continuous arrival of fresh OM to these shallow continental shelves (Buscail et al., 1990; Tselepides et al., 2000).

In contrast, surface sediments collected in a deeper and intensely-trawled flank of La Fonera Canyon, in the NW Mediterranean

margin, presented significantly lower turnover rates than the nearby untrawled grounds (Pusceddu et al., 2014). However, those sediment cores did not present signs of recently-accumulated sediment as observed in our sampling sites, further proving the dependence on the arrival of fresh and nutritionally-rich sediment in intensely trawled grounds to support benthic organisms living in these impacted deep-sea environments.

The high nutritional quality and OM turnover rates in recently-accumulated sediments from the trawled site suggest that high

OM fluxes in the Gulf of Castellammare could help bottom trawling grounds recover the nutritional characteristics of sedimentary OM. These results highlight that actions aimed at mitigating the impacts of bottom trawling should consider establishing temporary fishing closures, which would allow a longer-lived deposition of fresh OM on the seafloor, temporarily restoring sedimentary OM in trawling grounds. Further studies should be aimed in this direction to properly assess the effectiveness of establishing seasonal trawling closures as a management strategy to restore bottom trawling grounds.

## 5 Conclusion

Chronic and intense deep bottom trawling in the Gulf of Castellammare erodes large volumes of sediment, exposing over a century-old, compacted sediment that is depleted in OM. This continuous erosion limits the accumulation of fresh sediment, since any recently-deposited particulate matter is promptly removed due to the high trawling frequency. Nevertheless, the short-lived deposition of recent and nutritionally-rich organic matter leads to high turnover rates of labile OM. Our results emphasize that nutrient-deprived and eroded deep bottom trawling grounds are highly dependent on the arrival of fresh and nourishing particulate organic matter to sustain benthic communities, which can temporarily and partially abate the detrimental effects of bottom trawling in superficial sediment.

**Author contribution**

SP, AP, PP, and CL designed the scientific study. SP and CL retrieved the samples. SP and DM performed the analyses and TR processed the fishing effort data. SP wrote the manuscript. All authors contributed to the interpretation and discussion of the results, as well as the revision of the manuscript.

**Competing interests**

The authors declare that they have no conflict of interest.

**Acknowledgements**

The results presented in this study were obtained within the Exploring SiciLian CAnyoN Dynamics (ISLAND) Project, funded by FP7 EU Eurofleets2 (EU GA 312762), and the ABIDES Spanish Research Project (CTM2015-65142-R). Funding was provided to PM by the Generalitat de Catalunya (MERS 2017 SGR – 1588) and an Australian Research Council LIEF Project (LE170100219). We would like to thank the crew of the R/V Ángeles Alvariño and the ISLAND cruise team that helped collecting the samples. This work is contributing to the ICTA 'Unit of Excellence' (MinECo, MDM2015-0552). SP is supported by a predoctoral FPU grant from the Spanish government.

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





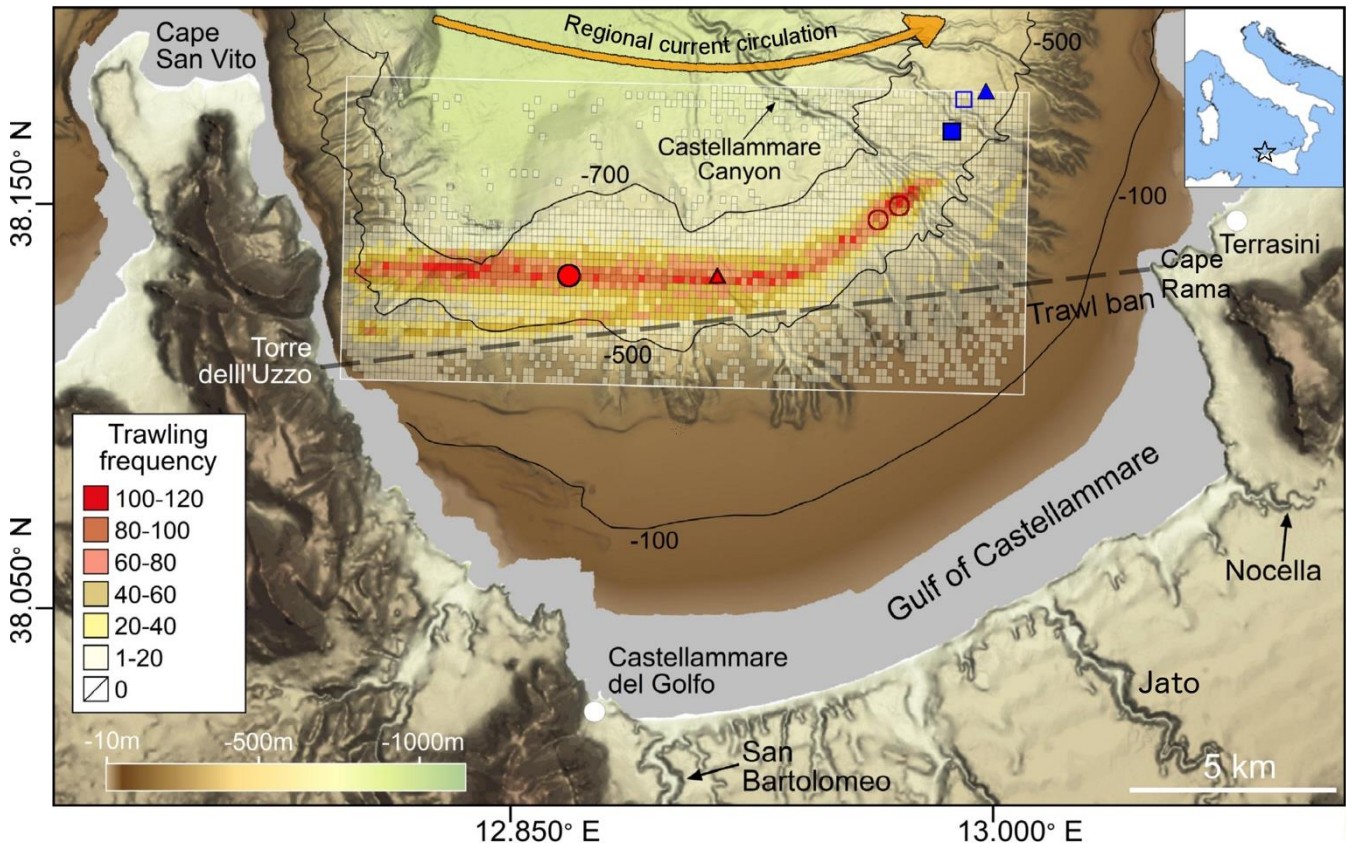

**Figure 1: Bathymetric map of the Gulf of Castellammare.** Location of the sediment cores sampled on the trawled (red circles) and untrawled (blue squares) sites. Unfilled sampling points indicate unsuccessful sediment core deployments. Seafloor images obtained from ROV dives are shown with triangles (see Fig. S1). The distribution of trawling grounds as trawling frequency (number of total hauls per grid area) in 2016 (January 1st to August 10th) was calculated for a 200 x 200 m grid. The limit of the trawl banned area between Torre dell'Uzzo and Cape Rama is indicated by a dashed line. Main trawling harbours (Castellammare del Golfo and Terrasini) and the most relevant ephemeral rivers (San Bartolomeo, Nocella, and Jato) are also annotated. The yellow arrow illustrates the direction of the regional surface current.





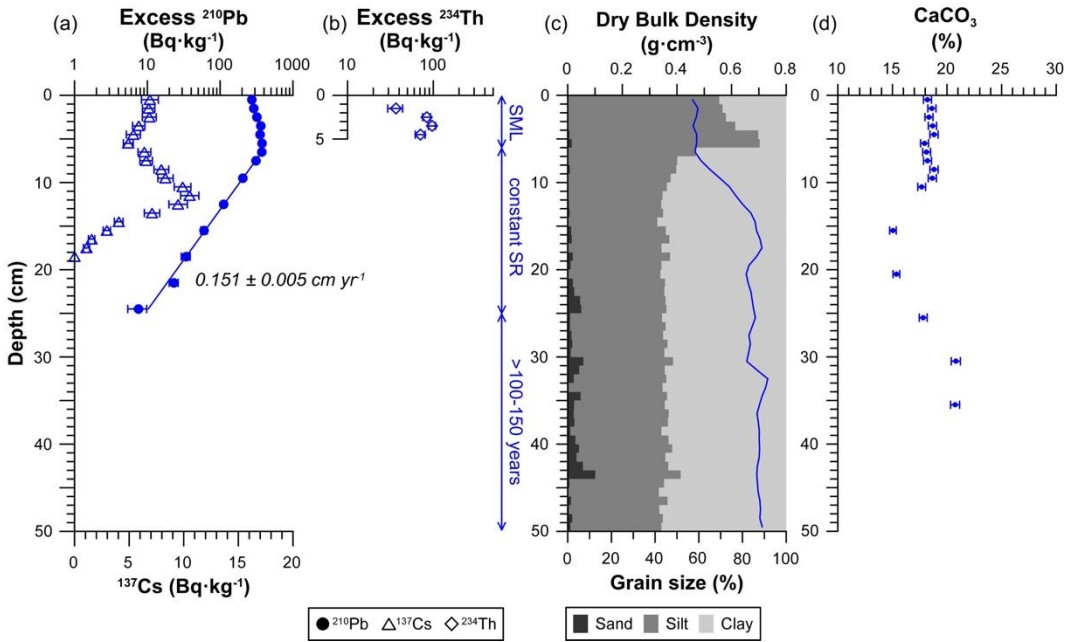

**Figure 2: Sedimentological and geochemical characteristics of the untrawled sediment core.** (a) Concentration profiles of excess $^{210}$Pb (circles), indicating the average sedimentation rate below the upper 6 cm, and $^{137}$Cs (triangles). (b) Concentration profile of excess $^{234}$Th in the upper 5 cm. (c) Dry bulk density and grain size. (d) CaCO$_3$ concentration profile. SML: Surface Mixed Layer; SR: Sedimentation Rate.

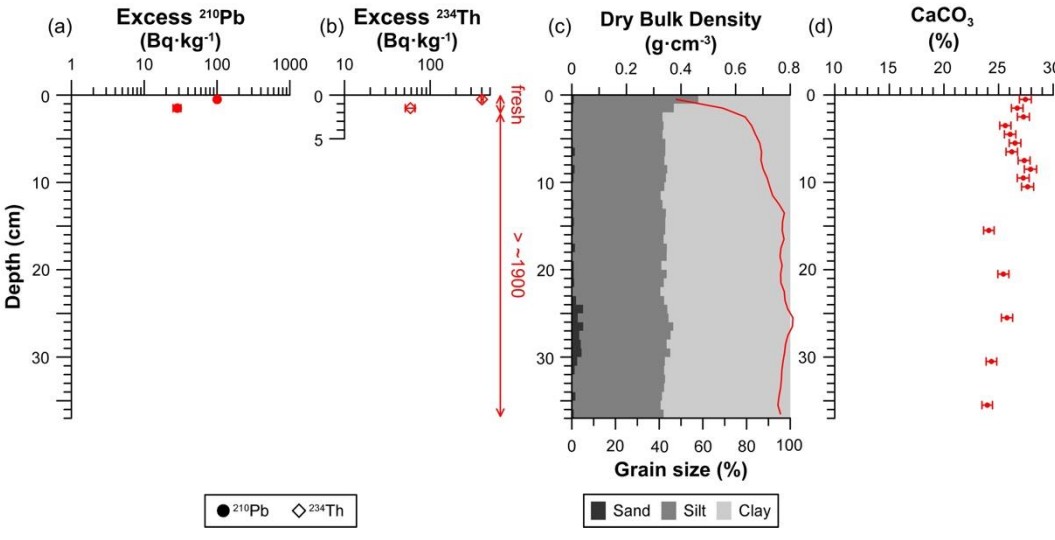

**Figure 3: Sedimentological and geochemical characteristics of the trawled sediment core.** (a) Concentration profiles of excess $^{210}$Pb. (b) Concentration profile of excess $^{234}$Th in the upper 5 cm. (c) Dry bulk density and grain size profile. (d) CaCO$_3$ concentration profile.



**Table 1: Summary of main parameters of the trawled and untrawled cores.** Data for grain size, $CaCO_3$, OC, TN, and OC/TN correspond to average values ± 1 standard deviation of that layer. MAR: mass accumulation rate, SAR: sediment accumulation rate.

| Core | Layer (cm) | Excess $^{234}$Th inventory (Bq·m$^{-2}$) | Excess $^{210}$Pb inventory (Bq·m$^{-2}$) | MAR (g·cm$^{-2}$·yr$^{-1}$) | SAR (cm·yr$^{-1}$) | Sand (> 63 μm) | Silt (4-63 μm) | Clay (< 4 μm) | CaCO₃ (%) | OC (%) | TN (%) | OC / TN |
|------|-----------|----------|----------|-----|-----|------|------|------|------|------|------|------|
| **Untrawled** | 0 – 6 | > 1080 ± 70 | 7310 ± 160 | Surface mixed layer | | < 1 | 77 ± 8 | 22 ± 8 | 18.4 ± 0.3 | 0.96 ± 0.02 | 0.133 ± 0.002 | 7.2 ± 0.2 |
| | 6 – 25 | - | 10480 ± 490 | 0.090 ± 0.003 | 0.151 ± 0.005 | 1.5 ± 0.4 | 44 ± 5 | 54 ± 4 | 17 ± 2 | 0.93 ± 0.03 | 0.132 ± 0.004 | 7.1 ± 0.2 |
| | 25 – 50 | - | - | * | | 3 ± 2 | 42 ± 2 | 55 ± 2 | 20 ± 2 | 0.79 ± 0.06 | 0.110 ± 0.010 | 7.2 ± 0.4 |
| **Trawled** | 0 – 2 | 1000 ± 50 | 340 ± 30 | Fresh sediment | | < 1 | 52 ± 8 | 48 ± 8 | 27.1 ± 0.5 | 0.79 ± 0.02 | 0.101 ± 0.002 | 7.8 ± 0.4 |
| | 2 – 37 | - | - | * | | 1.4 ± 1.5 | 41.4 ± 1.1 | 57.3 ± 1.3 | 26.1 ± 1.3 | 0.73 ± 0.03 | 0.103 ± 0.007 | 7.1 ± 0.4 |

* Section depleted of excess $^{210}$Pb, corresponding to sediment deposited before ~1900.



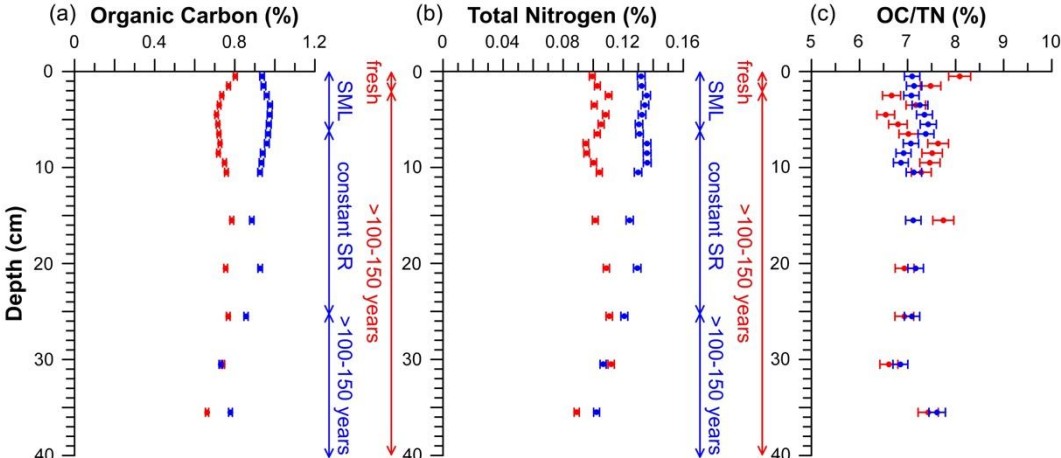

**Figure 4: Profiles from elemental analyses of untrawled (blue) and trawled (red) cores.** Organic carbon **(a)**, total nitrogen **(b)**, and the OC/TN ratio **(c)**. The analytical error is represented by error bars.

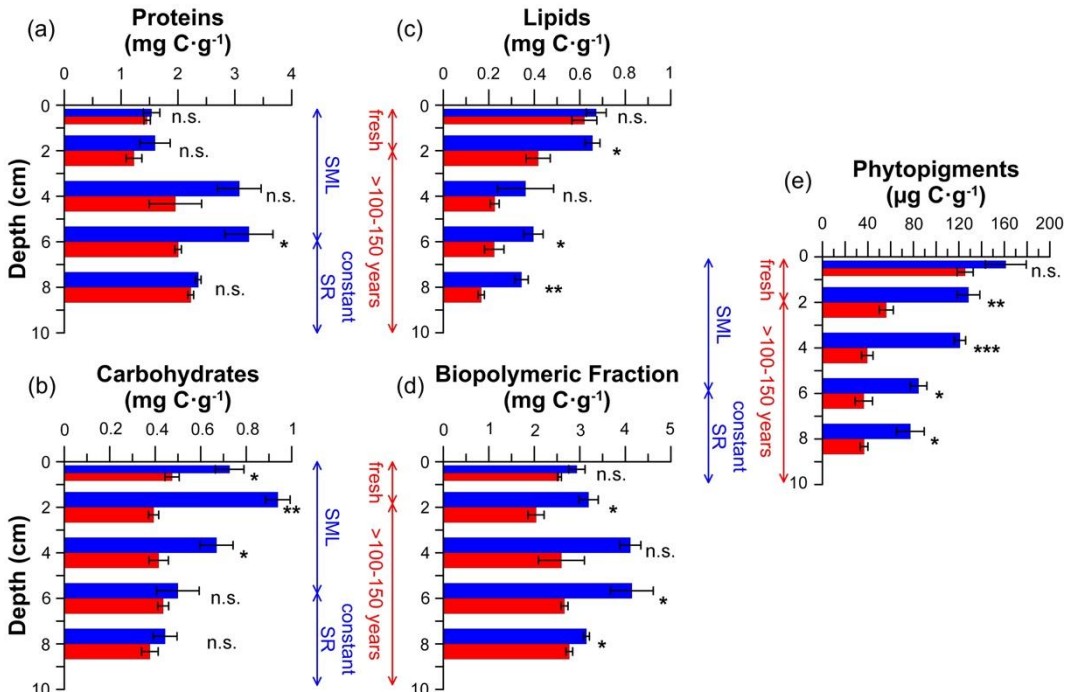

**Figure 5: Organic matter quantity of untrawled (blue) and trawled (red) sites.** Proteins **(a)**, carbohydrates **(b)**, lipids **(c)**, biopolymeric carbon **(d)**, and phytopigments **(e)**. Asterisks next to bars denote significant difference of post-hoc permutational pairwise tests between trawled and untrawled sites: * = $p < 0.05$; ** = $p < 0.01$; *** = $p < 0.001$; n.s.= not significant.



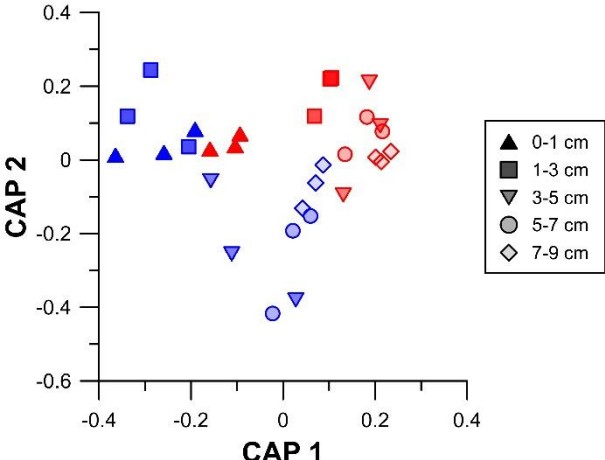

**Figure 6: Variations in the biochemical composition of the sedimentary organic matter.** Bi-plot after canonical analysis of the principal coordinates. Note that symbols represent the same core depth for both trawled (red) and untrawled (blue) sites, and that increasing depth is also illustrated by a fading filling.

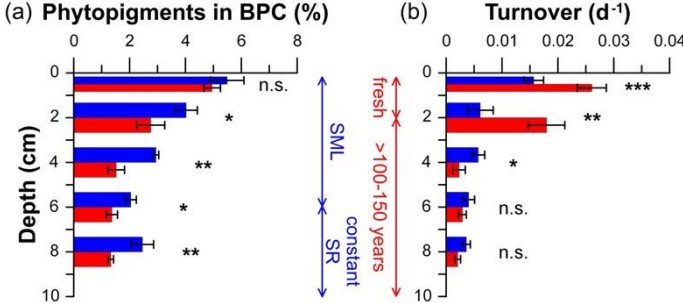

**Figure 7: Organic matter freshness and degradation rates of trawled (red) and untrawled (blue) cores.** Relative contribution of phytopigments to biopolymeric C **(a)** and protein turnover rates **(b)**. Asterisks next to bars denote significance of post-hoc permutational pairwise tests between trawled and untrawled sites: * = $p < 0.05$; ** = $p < 0.01$; *** = $p < 0.001$; n.s.= not significant.