# Peer review of "Effects of the arrival of fresh organic matter on eroded and nutrient-depleted trawling grounds (Gulf of Castellammare, SW Mediterranean)"

_Biogeosciences, 2019_

## Referee Comment (RC1) · Sebastiaan van de Velde (Referee) · 5 Jul 2019

Paradis and coworkers describe the impact of bottom trawling on the organic matter content and quality in marine sediments. They do this by measuring the organic carbon quantity, as well as proxies for organic matter quality (like pigments, proteins, lipids . . .) in sediment cores collected at two sites with a different trawling intensity. They show that both sites experienced different intensities of trawling by measuring measuring radionuclides (210Pb, 137Cs and 234Th).

This paper discusses an important topic about which very little is currently known. Bottom trawling fishing represents likely one of the dominant anthropogenic impacts on coastal and shelf sediments, and is rapidly expanding in deeper waters. This represents a potential (and largely unknown) threat for the marine environment, which is already under a lot of pressure. This study has been well designed and executed, and the data presented is of high quality and will be of interest to many readers, and possibly also to the general public. I read it with great interest and offer only a few comments/suggestions for consideration along with some minor editorial suggestions.

Comments:

1. I feel that the title does not really reflect the contents. Most of the paper deals with the impact of bottom trawling fishing on the sedimentary organic matter dynamics, of which the arrival of fresh organic matter is part. Perhaps something like 'the impact of trawling on organic matter dynamics in sediments of the Gulf of Castellammare (SW Mediterranean)' would be more appropriate.

2. I like that the paper is written very concise and to the point. I do however feel that the abstract is a bit out of balance with the rest of the paper. If possible, I would try to shorten it somewhat.

3. Introduction: It might be relevant to mention the common depth range of the sediment that is resuspended after trawling, as it might be important when considering the impact of changing the frequency of trawling. For example, if your site has a sedimentation rate of 0.15 yr-1, and you decrease trawling frequency to once every 10 years, and the trawl resuspends the upper 15 cm of sediment, the impact is still considerable.

4. P10L14: Would you expect coarser sediment to be washed away following trawling induced resuspension? I would assume the inverse happens, that the finer-grained, OM rich material gets washed away, and the coarser material remains behind. Also, the grain size of the trawled site has near-identical values for its grain size. Unless you mean that the finest material does not settle as fast as the intermediate grain sizes (but

that should then be explained a bit clearer ).

5. P10L22-26: Another explanation could be that the resuspension and mixing stimulates the breakdown of organic matter that is already present, thus leading to lower concentrations. This is also what you allude to in P11L15-23, an increase in mineralization rates due to the mixing of refractory and fresh compounds. If it is noticeable after the deposition of fresh material, I assume it will play a role over the longer term too. Most likely the truth exists somewhere in the middle, and the impact of bottom trawling induced resuspension events is likely very dependent on the exact grain size. For example, we found recently that this mixing stimulates organic matter breakdown in a fine grained (63% clay) sediment in the North Sea (van de Velde et al., 2018, Scientific Reports), whereas Tiano et al. (2019, ICES Journal of Marine Science) found that sediment metabolism actually went down after trawling. The North Sea site of Tiano et al. (which is a reference you should include) was much however much sandier (30% silt, 40% fine sand). Maybe it would be worth expanding the discussion a bit, and considering both end-member scenarios.

6. P11L29: this is a nice section to end the discussion, maybe it would be worth expanding this a bit, maybe by being a bit more concrete in the potential mitigation effects? This could be related to my comment about the depth of the bottom trawling, how long would temporary need to be to really mitigate the effect?

Minor editorial suggestions:

As a more general remark, you say you sliced cores in triplicate, but I only see one profile per figure and per site (and the captions says that the error bars represent the analytical error). What happened to the other 2 cores that were sliced?

P4L13-15: 'concentrations of sedimentary organic matter in superficial sediments tend to increase' and later 'stimulate mineralization of buried and refractory organic matter' This seems contradictory to me, as stimulating mineralization would decrease organic matter.

P3L17: move 'are' between 'sedimentary organic matter' and 'by'

P4L13: you mention that you slice cores on deck, up to 9cm depth, but later you show figures with date up to 20 cm depth? (e.g. Fig.2)

P5L11: why did you limit measurements to the upper 5 cm? I assume that is because activities dropped below the detection limit, but it might be nice to mention that here.

P8L8: Maybe it is not relevant for this paper, but why would the CaCO3 contents differ?

P11L18: Aller (1994) is not an appropriate reference, this paper deals with bioturbation an redox oscillations, not self-priming.

Figs 2 and 3 could be combined into 1

Refs:

van de Velde S., Van Lancker V., Hidalgo-Martinez S., Berelson W.M. and Meysman F.J.R. (2018) Anthopogenic disturbance keeps the coastal seafloor biogeochemistry in a transient state. Scientific Reports. DOI:10.1038/s41598-018-23925-y

Tiano J.C., Witbaard R., Bergman M.J.N., van Rijswijk P., Tramper A., van Oevelen D. and Soetaert K. (2019) Acute impacts of bottom trawl gears on benthic metabolism and nutrient cycling. ICES Journal of Marine Science. DOI:10.1093/icesjms/fsz060

---

## Referee Comment (RC2) · Xavier Durrieu de Madron (Referee) · 30 Jul 2019

In this paper, the authors attempt to highlight the long-term (over decades) impact of bottom trawling on the sedimentological, radioisotopic and biogeochemical characteristics of the sediment by comparing two contrasting sites, one regularly impacted by trawling activity and the other outside the area heavily impacted by trawling. They thus reveal important differences that demonstrate the erosive effect of recurrent trawling activity, the transport of part of the resuspended sediment to non-trawled areas, and the ability of benthic communities to adapt to substrate alteration and organic matter

inputs of planktonic origin. The manuscript is generally concise and clear. It is also well organized and illustrated. The following remarks and suggestions should clarify some points.

Page 2, Lines18-19. The comparison of the impact of storms versus trawling is not discussed in the article by Durrieu de Madron et al, 2005, but by Ferré et al, 2008. (Ferré B., X. Durrieu de Madron, C. Estournel, C. Ulses, G. Le Corre (2008). Impact of natural and anthropogenic (trawl) resuspension on the export of particulate matter to the open ocean. Application to the Gulf of Lion (NW Mediterranean). Continental Shelf Research, 28, 2071–2091).

Page 3, Lines 6-8 Since when has bottom trawling been practiced on the continental slope? Is it since the 1990 ban or was this area trawled before? This information would be useful to give an effective duration of trawling activity in the study area.

Page 2, Line 26. It is a cyclonic circulation (anti-clockwise) and not an anticyclonic circulation. On the other hand, I imagine that currents on the continental shelf are variable and strongly impacted by wind, while the circulation along the continental slope is probably more permanent. I suggest simply writing "A cyclonic along-slope current dominates the Gulf's circulation".

Page 3, Lins 12-13. The sampling strategy includes three multi-tube corer deployments at the same station from which 3 cores are collected. Did you analyze each slice of sediment of the 9 cores thus collected and then estimate the mean and standard deviations, or did you mix all the sedimentary material of the different cores before analyzing it and the error bars shown correspond then to the instrumental error.

Page 3, Line 21-24. Can you indicate the size limits between clays and silts, and silts and sands?

Page 4, Line 2. Indicate the maximum depth of the cores on which these analyses were performed.

[Figure]

Page 7, Line 8-13. It would be useful here and for the discussion to know more about the fishing gears. Can you specify the main types and characteristics of bottom trawls used by fishermen in this region? Are they beam or otter trawls? Are they equipped with rollers or chains?

Page 10, Lines 14-16. Do you have any information on the intensity of the bottom current to estimate their capacity to transport or even remobilize fine sediment?

Page 10, Lines 29-34. Do you think that the benthic and epi-benthic communities are the same between the two sites (trawled and untraveled) given the differences in the substrate? Could different species induce significant differences in the organic matter turnover rate? Meiofauna biodiversity is not addressed in this article, but I think it would be interesting to consider this possibility in the discussion (if it makes sense)?

Captions of Figures 2, 3, 4, 5 and 7. Explain the vertical blue and red scales, as well as acronyms (SML: Surface Mixed Layer, constant SR : constant Sedimentation Rate)

---

## Author Comment (AC1) · 29 Sep 2019

We would like to thank Dr. Van de Velde for the time taken to read and review our manuscript, and for raising relevant aspects that need clarification in the text. Please find below our detailed responses to his comments.

1. I feel that the title does not really reflect the contents. Most of the paper deals with the impact of bottom trawling fishing on the sedimentary organic matter dynamics, of which the arrival of fresh organic matter is part. Perhaps something like 'the impact of

trawling on organic matter dynamics in sediments of the Gulf of Castellammare (SW Mediterranean)' would be more appropriate.

RESPONSE: We believe that the title emphasizes the novelty of our manuscript, since previous papers have already addressed the impacts of bottom trawling in sediment erosion and depletion of sedimentary organic matter, but not on the recent deposition of fresh sediment. Nevertheless, we have modified the title to "Organic matter contents and degradation in a highly trawled area during fresh particle inputs (Gulf of Castellammare, SW Mediterranean)" to englobe the different aspects studied in our manuscript.

2. I like that the paper is written very concisely and to the point. I do however feel that the abstract is a bit out of balance with the rest of the paper. If possible, I would try to shorten it somewhat.

RESPONSE: The abstract has been condensed in the revised manuscript.

3. Introduction: It might be relevant to mention the common depth range of the sediment that is resuspended after trawling, as it might be important when considering the impact of changing the frequency of trawling. For example, if your site has a sedimentation rate of 0.15 cm/yr, and you decrease trawling frequency to once every 10 years, and the trawl resuspends the upper 15 cm of sediment, the impact is still considerable.

RESPONSE: Indeed, knowing the depth-range of sediment that is resuspended by bottom trawlers would be crucial to understand the vulnerability of our study site to bottom trawling activities and establish efficient management strategies. There have been studies that model the amount of sediment resuspended by bottom trawlers, which indicate that type of trawling gear, sediment grain size, and hydrodynamic drag exerted by the trawling gear influence the mobilization of sediment (see O'Neill and Ivanovic, 2016, ICES Journal of Marine Systems; O'Neill and Summerbell, 2016, Journal of Marine Systems). These studies highlight that penetration depth of trawling gear on the seafloor and sediment resuspension do not always present an evident positive correlation, since some of this sediment is simply overturned and/or displaced laterally.

Hence, providing a penetration depth of bottom trawling gear would not be indicative of the depth-range of sediment being resuspended, either. With our data, we cannot provide information on the amount of sediment being eroded per trawler. However, we have sufficient evidence, based on our results, that the overall erosion rate of trawlers is greater than the sedimentation rate in the Gulf of Castellammare. This is inferred due to the coincident penetration depths of both excess Th-234 and excess Pb-210, which have considerably different half-lives, indicating that the upper 2 cm of sediment was recently deposited, whereas sediment below these sections had been deposited more than a century prior to sampling. This highlights the vulnerability of bottom trawling in deep environments, where the sedimentation rate is lower than shallower continental shelves. Please see the response to your comment 6, which also deals on management strategies to mitigate this impact.

4. P10L14: Would you expect coarser sediment to be washed away following trawling induced resuspension? I would assume the inverse happens, that the finer-grained, OM rich material gets washed away, and the coarser material remains behind. Also, the grain size of the trawled site has near-identical values for its grain size. Unless you mean that the finest material does not settle as fast as the intermediate grain sizes (but that should then be explained a bit clearer).

RESPONSE: Indeed, we should observe the advection of fine sediment, and not coarse sediment, with a preferential deposition based on particle size. The untrawled core was collected approximately 1 km downcurrent from trawling grounds, where silty sediment will be preferentially deposited in comparison to finer sediment such as clay particles. Clay sediment, however, can remain in suspension and travel greater distances, eventually redepositing farther away. For instance, a study on the distribution of trawling-induced resuspension of sediment in the Koster Sea on the west coast of Sweden observed that silt particles can travel up to 7 km from trawling grounds, whereas finer clay particles can travel beyond 28 km (Linders et al., 2018, ICES Journal of Marine Systems). We acknowledge that this message may not have been clear

due to poor word choice, referring to "silt" particles as "coarse" sediment. The amended manuscript now reads: "Provided the high capacity of bottom trawling gear to resuspend sediments (Martin et al., 2014a, 2014c; Oberle et al., 2018; Puig et al., 2012), the siltation of superficial sediments on the untrawled site could probably be explained by the preferential deposition of siltier particles resuspended from an adjacent trawling ground located ~1 km up-current from this sampled site (Fig.1). Finer clay particles resuspended by bottom trawlers can be advected at farther distances along the margin (Linders et al., 2018)."

5. P10L22-26: Another explanation could be that the resuspension and mixing stimulate the breakdown of organic matter that is already present, thus leading to lower concentrations. This is also what you allude to in P11L15-23, an increase in mineralization rates due to the mixing of refractory and fresh compounds. If it is noticeable after the deposition of fresh material, I assume it will play a role over the longer term too. Most likely the truth exists somewhere in the middle, and the impact of bottom trawling induced resuspension events is likely very dependent on the exact grain size. For example, we found recently that this mixing stimulates organic matter breakdown in a fine grained (63% clay) sediment in the North Sea (van de Velde et al., 2018, Scientific Reports), whereas Tiano et al. (2019, ICES Journal of Marine Science) found that sediment metabolism actually went down after trawling. The North Sea site of Tiano et al. (which is a reference you should include) was much however much sandier (30% silt, 40% fine sand). Maybe it would be worth expanding the discussion a bit, and considering both end-member scenarios.

RESPONSE: Diverse results have been found on the fate of OM in trawling grounds, some of them summarized in Martin et al., 2014, Anthropocene. As the reviewer mentions, one of the reasons of these contradictory results could be the type of sediment in each environment: whether trawling occurs on non-cohesive sandy seafloors or on cohesive muddy seafloors. We limited our study to comparing the effects of bottom trawling to other cohesive environments (including the results portrayed in van de

[Figure]

Velde et al., 2018, Scientific Reports), and merely distinguishing between biogeochemical impacts in shallower and deeper trawling grounds. We agree with the reviewer that it would be worth exploring the influence of grain size, but it is not the scope of this paper. We believe that a proper review of the impacts of bottom trawling on the biogeochemistry of OM should be done to assess the influence of grain size, water depth, and other factors. Nevertheless, we have included the following sentences in the Introduction of the amended manuscript, which highlight the broad range of effects that bottom trawling can generate in different environments: "The effects that these perturbations generate on sedimentary OM can vary in cohesive (i.e. muddy sediment with high clay content) and non-cohesive (i.e. sandy seafloor) sediment. For instance, trawling on cohesive sediments can increase superficial concentrations of sedimentary OM (Palanques et al., 2014; Pusceddu et al., 2005a; Sciberras et al., 2016; Polymenakou et al., 2005), whereas trawling on coarse non-cohesive sediments can exert null or minimal effects on OM contents and benthic community metabolism (Hale et al., 2017; Tiano et al., 2019; Trimmer et al., 2005)."

6. P11L29: this is a nice section to end the discussion, maybe it would be worth expanding this a bit, maybe by being a bit more concrete in the potential mitigation effects? This could be related to my comment about the depth of the bottom trawling, how long would temporary need to be to really mitigate the effect?

RESPONSE: Our study suggests that the ephemeral deposition of fresh and nutritious sediment could be sustaining the otherwise starved benthic communities. Establishing temporal trawling closures would allow a longer-lived deposition of fresh sediment, temporarily restoring sedimentary OM in trawling grounds which could be beneficial to the benthic communities inhabiting this area. However, with the punctual information we have of the impacts of bottom trawling and the effect of the deposition of fresh sediment, we can not provide additional details of how a temporal trawling closure should be implemented (i.e. length or season of the trawling closure). Hence, we believe that further studies assessing the viability of these mitigation practices should be carried out. From the sedimentological perspective, a temporal trawling closure would not solve the issue of erosion in trawling grounds. For instance, assuming a regional sedimentation rate of 0.09 g/cm2/yr, or 0.15 cm/yr, a trawling closure of a decade would allow the accumulation of 0.9 g/cm2, or 1.5 cm of "new" sediment. Such a long trawling closure isn't feasible from a socioeconomic perspective, nor efficient. To solve this problem, other management strategies that reduce the rate of erosion would need to be studied, such as reducing the trawling frequency or changing their trawling gear to minimize sediment remobilization. This additional issue has been addressed in the revised manuscript by modifying the discussion's closing paragraph from the original manuscript to the following: "These results confirm that actions aimed at mitigating the impacts of bottom trawling include the implementation of temporary fishing closures, allowing for a longer-lived deposition of fresh OM on the seafloor. However, such temporary trawling closures would most probably not allow the full restoration of fresh sediment from trawl-induced erosion, given the low sedimentation rates found on these deep environments. Further management strategies would need to be implemented to mitigate the impacts of bottom trawling erosion (Depestele et al., 2019), which would magnify the effect of temporary closures on the restoration of sedimentary OM in nutrient-deprived trawling grounds."

Minor editorial suggestions: As a more general remark, you say you sliced cores in triplicate, but I only see one profile per figure and per site (and the captions says that the error bars represent the analytical error). What happened to the other 2 cores that were sliced?

RESPONSE: Triplicate cores were taken only for organic matter analyses (protein, carbohydrate, lipid, phytopigment, turnover rates), whereas the remaining analyses (dry bulk density, grain size, radiochemical analyses, and elemental analyses) were carried out in one sediment core from each site. We specified that error bars in Fig. 4, for instance, represent analytical errors, and we should have also specified that Figs. 5 and 7 represent mean and standard errors of triplicate samples. This has been clarified

in the revised manuscript.

P2L13-15: 'concentrations of sedimentary organic matter in superficial sediments tend to increase' and later 'stimulate mineralization of buried and refractory organic matter' This seems contradictory to me, as stimulating mineralization would decrease organic matter.

RESPONSE: Indeed, this would seem contradictory and should be clarified. High OM concentrations initially lead to high remineralization, which eventually lower sedimentary OM concentrations. It is basically an issue of the time-scale of these processes. In Polymenakou et al., 2005, Continental Shelf Research and in Pusceddu et al., 2005, Continental Shelf Research, the onset of trawling activities initially led to higher sedimentary organic matter concentrations, possibly due to mixing, which was accompanied with higher OM degradation. However, re-sampling of these trawled sites a few months later indicated a decrease of sedimentary organic matter concentrations, attributed to the high degradation rates observed earlier. Nevertheless, this apparent contradictory sentence was removed to avoid confusions. See the response to your comment 5 for how this issue was clarified.

P3L17: move 'are' between 'sedimentary organic matter' and 'by'

RESPONSE: This grammatical correction has been included in the revised manuscript.

P4L13: you mention that you slice cores on deck, up to 9cm depth, but later you show figures with date up to 20 cm depth? (e.g. Fig.2)

RESPONSE: We sliced triplicate cores intended for organic matter analyses up to 9 cm, whereas the remaining cores, used to analyse the remaining parameters (dry bulk density, grain size fraction, radiochemical analyses), were completely sliced in 1 cm intervals.

P5L11: why did you limit measurements to the upper 5 cm? I assume that is because activities dropped below the detection limit, but it might be nice to mention that here.

[Figure]

RESPONSE: Only the upper 5 cm were analysed for excess Th-234 since samples need to be measured within two half-lives (approximately 6 weeks), after which 75 % of excess Th-234 would have decayed, rendering its quantification unreliable and with a high uncertainty. Since we didn't find detectable concentrations of Cs-137 in the trawled site in these upper 5 cm, we considered unnecessary to analyse deeper samples of this core, whereas gamma measurements for Cs-137 in the untrawled core were conducted for deeper layers until concentrations were below detection limit. This has been clarified in the revised manuscript.

P8L8: Maybe it is not relevant for this paper, but why would the CaCO3 contents differ?

RESPONSE: Trawled sites presented higher ($\sim$27 %) CaCO3 concentrations than the untrawled site ($\sim$17 %) in the upper 10 cm, although the difference becomes smaller at deeper sections in the cores. This could be related to differences of (or proximity to) riverine sediment sources, or to a higher presence of broken shells or foraminifera. With the available information, we cannot explain the exact reasons for this phenomenon and we prefer not to speculate about this aspect in the paper.

P11L18: Aller (1994) is not an appropriate reference, this paper deals with bioturbation and redox oscillations, not self-priming.

RESPONSE: This reference was accordingly removed in the revised manuscript.

Figs 2 and 3 could be combined into 1

RESPONSE: We would rather keep these two figures separate, since combining them would make the figure too dense.

---

## Author Comment (AC2) · 29 Sep 2019

We would like to thank Dr. Durrieu de Madron for the time taken to read and review our manuscript. Our detailed responses to his remarks and how they were addressed in the revised manuscript are provided below:

Page 2, Lines18-19. The comparison of the impact of storms versus trawling is not discussed in the article by Durrieu de Madron et al, 2005, but by Ferré et al, 2008. (Ferré B., X. Durrieu de Madron, C. Estournel, C. Ulses, G. Le Corre (2008). Impact of

natural and anthropogenic (trawl) resuspension on the export of particulate matter to the open ocean. Application to the Gulf of Lion (NW Mediterranean). Continental Shelf Research, 28, 2071–2091).

RESPONSE: The reference has been corrected in the amended manuscript.

Page 3, Lines 6-8 Since when has bottom trawling been practiced on the continental slope? Is it since the 1990 ban or was this area trawled before? This information would be useful to give an effective duration of trawling activity in the study area.

RESPONSE: Intense bottom trawling activities in the Castellammare region has been practiced for decades prior to the banning. The following sentence has been included in the revised manuscript: "First data of bottom trawlers in the area go back to the 1960s, but this fishery became more active since the 1980s (European Comission Fisheries & Maritime Affairs, 2014), as a result of the modernization of the Sicilian trawling fleet (L.R. 1/1980, L.R. 26/1987)."

Page 2, Line 26. It is a cyclonic circulation (anti-clockwise) and not an anticyclonic circulation. On the other hand, I imagine that currents on the continental shelf are variable and strongly impacted by wind, while the circulation along the continental slope is probably more permanent. I suggest simply writing "A cyclonic along-slope current dominates the Gulf's circulation".

RESPONSE: This mistake has been corrected in the amended manuscript.

Page 3, Lines 12-13. The sampling strategy includes three multi-tube corer deployments at the same station from which 3 cores are collected. Did you analyze each slice of sediment of the 9 cores thus collected and then estimate the mean and standard deviations, or did you mix all the sedimentary material of the different cores before analyzing it and the error bars shown correspond then to the instrumental error.

RESPONSE: Three sediment cores from triplicate multicorer deployments were retrieved at each station for organic matter analyses (proteins, carbohydrates, lipids, phytopigment, and turnover rate analyses). These analyses were conducted for each slice of the 9 sediment cores. The mean and standard errors of each sampled section was calculated for each depth at both sites (trawled and untrawled) and these results are presented in Figs. 5 and 7. On the other hand, a single sediment core from one of the three deployments was used to analyse the remaining parameters (sediment dry bulk density, grain size, radiochemical analyses). For these analyses, the error bars of Figs. 2-4 correspond to their analytical error. This has been clarified in the Figure captions of the revised manuscript.

Page 3, Line 21-24. Can you indicate the size limits between clays and silts, and silts and sands?

RESPONSE: The size limits between clays (< 4 $\mu$m), silts (4-63 $\mu$m), and sands (> 63 $\mu$m) were given in Table 1. However, they have also been included in-text under Sect. 2.3.

Page 4, Line 2. Indicate the maximum depth of the cores on which these analyses were performed.

RESPONSE: Pb-210 analyses were conducted downcore until 37 cm and 49 cm for the trawled and untrawled site, respectively. This information has been included in the revised manuscript.

Page 7, Line 8-13. It would be useful here and for the discussion to know more about the fishing gears. Can you specify the main types and characteristics of bottom trawls used by fishermen in this region? Are they beam or otter trawls? Are they equipped with rollers or chains?

RESPONSE: Bottom trawling in the Gulf of Castellammare is mainly conducted by bottom otter trawls. The following sentence has been included in the revised manuscript: "Bottom trawlers in this gulf operate using otter trawl gear, a trawling technique which consist of dragging a wide net that is held open and in contact with the seafloor by two

otter doors (Martín et al., 2014a)."

Page 10, Lines 14-16. Do you have any information on the intensity of the bottom current to estimate their capacity to transport or even remobilize fine sediment?

RESPONSE: Bottom current in the upper shelf of the Gulf of Castellammare has an average speed of 0.1-0.2 m/s, but can sometimes reach 0.4 m/s (Sarà et al., 2006). Unfortunately, there is no data of bottom currents on the slope close to our sampling sites, but assuming similar bottom current intensities as those observed on the shelf, it wouldn't cause enough shear stress to remobilize the fine-grained cohesive sediment of our study site. This information has been included both in 2.1 Study area and in the aforementioned section.

Page 10, Lines 29-34. Do you think that the benthic and epi-benthic communities are the same between the two sites (trawled and untrawled) given the differences in the substrate? Could different species induce significant differences in the organic matter turnover rate? Meiofauna biodiversity is not addressed in this article, but I think it would be interesting to consider this possibility in the discussion (if it makes sense)?

RESPONSE: Turnover rates were calculated from extracellular enzymatic activities produced by bacteria, hence, the turnover rates presented in our manuscript don't reflect metazoan consumption of organic matter. Nevertheless, trawling will undoubtedly cause differences not only in sedimentary organic matter, but also in epi-benthic communities, as observed in deep bottom trawling grounds off the NW Mediterranean (Pusceddu et al., 2014, PNAS). A separate paper dealing with epi-benthic community in sediment cores collected during the ISLAND cruise is under development. This under-construction paper will partly deal with the effects of bottom trawling, using our current manuscript as reference of the physical impacts of bottom trawling and its effect in organic matter content and degradation in the Gulf of Castellammare.

Captions of Figures 2, 3, 4, 5 and 7. Explain the vertical blue and red scales, as well as acronyms (SML: Surface Mixed Layer, constant SR : constant Sedimentation Rate)

[Figure]

RESPONSE: The vertical annotations have been explained in the figure caption of the revised manuscript.